# Anti-Cancer Properties of Coix Seed Oil against HT-29 Colon Cells through Regulation of the PI3K/AKT Signaling Pathway

**DOI:** 10.3390/foods10112833

**Published:** 2021-11-17

**Authors:** Chunlei Ni, Bailiang Li, Yangyue Ding, Yue Wu, Qiuye Wang, Jiarong Wang, Jianjun Cheng

**Affiliations:** College of Food Science, Northeast Agricultural University, Harbin 150030, China; nichunlei@neau.edu.cn (C.N.); 15846092362@163.com (B.L.); dingyangyue77@163.com (Y.D.); wuyue2020@neau.edu.cn (Y.W.); wqye997@163.com (Q.W.); wjrsci@163.com (J.W.)

**Keywords:** coix seed oil, anti-colon cancer, cell cycle, apoptosis, PI3K/AKT

## Abstract

This study aims to observe the effects of coix seed oil (CSO) on HT-29 cells and investigate its possible regulation mechanism of the PI3K/Akt signaling pathway. Fatty acid analysis showed that coix seed oil mainly contains oleic acid (50.54%), linoleic acid (33.76%), palmitic acid (11.74%), and stearic acid (2.45%). Fourier transform infrared results found that the fatty acid functional groups present in the oil matched well with the vegetable oil band. The results from CCK-8 assays showed that CSO dose-dependently and time-dependently inhibited the viability of HT-29 cells in vitro. CSO inhibited cell viability, with IC_50_ values of 5.30 mg/mL for HT-29 obtained after 24 h treatment. Morphological changes were observed by apoptotic body/cell nucleus DNA (Hoechst 33258) staining using inverted and fluorescence microscopy. Moreover, flow cytometry analysis was used to evaluate the cell cycle and cell apoptosis. It showed that CSO induced cell apoptosis and cycle arrest in the G_2_ phase. Quantitative real-time PCR and Western blotting revealed that CSO induced cell apoptosis by downregulating the PI3K/AKT signaling pathway. Additionally, CSO can cause apoptosis in cancer cells by activating caspase-3, up-regulating Bax, and down-regulating Bcl-2. In conclusion, the results revealed that CSO induced G_2_ arrest and apoptosis of HT-29 cells by regulating the PI3K/AKT signaling pathway.

## 1. Introduction

Colorectal cancer is the third most common cancer in the world, and it is also one of the most deadly cancers [1]. Traditional treatment methods have certain side effects [2]. Natural plant products have attracted increasing attention in cancer treatment due to their low side effects and high efficacy [3]. For instance, dark sweet cherry phenolics, curcumin, isoflavonoids, resveratrol, epigallocatechin gallate (EGCG), tea tree oil extract, corn oil, and isoliquiritigenin (ISL) can be used for cancer therapy and to help reverse the drug resistance of certain cancer cells [4,5,6,7,8,9,10,11]. Accordingly, many works have explored the protective effect of natural sources to inhibit cancer cells.

In cereals, coix seed (Coix lachryma-jobi L.) has attracted wide attention due to its various biological activities. It is a Chinese herbal plant that is planted in Asia, Africa, and the marginal regions of the Mediterranean. Coix seed consists of a series of biologically active ingredients, such as polysaccharides, proteins, polyphenols, oils, coixenolide, sterols, and others [12,13]. In particular, coix seed oil (CSO) has key anti-tumor effects, neutralizing free radicals, and also anti-cancer, anti-inflammatory, and anti-aging aspects [13,14,15,16,17]. As the major bioactive ingredient identified in Coix seeds, fatty acids have been reported to present significant anti-tumor activity [18]. In coix seed oil, unsaturated fatty acid residues in all triglyceride fatty acid residues are over 84%, including mainly oleic acid (31.42%) and linoleic acid (47.38%) [19,20]. Coix seed oil exhibited anti-cancer activity in T24 cancer cell lines; this was attributed to palmitic acid and linoleic acid in an appropriate ratio to oleic acid [21]. CSO inhibits fatty acid synthase (FAS) and can be used for the treatment of tumors [20]. Previous studies have shown that CSO can intercept the cell cycle in the G2 phase, reduce cell mitosis, and inhibit cancer cell proliferation. It is used in the treatment of primary malignancies, including lung, liver, gastric, and breast cancers, because of its anti-proliferative and pro-apoptotic effects on many tumors in vitro and in vivo [22,23,24,25].

As the main signaling pathway in cancer research, the PI3K/AKT pathway has attracted the attention of many researchers. The PI3K/AKT pathway exerts many effects in different cancer progressions related to the apoptosis, autophagy, and survival of cells [26,27,28]. In many types of cancer, the phosphoinositotide 3-kinase (PI3K) pathway shows hyperactivity due to mutations, amplification, deletions, methylation, and post-translational modifications, resulting in reduced apoptosis and sustained proliferation. AKT proteins are activated by the PI3K pathway to improve cell cycle progression, which, in turn, drives many cellular functions. However, there are relatively few studies on the signaling pathway and on the promotion of apoptosis of colon cancer cells by CSO. Therefore, it is of great significance to explore the role of CSO in the regulation of the PI3K/AKT signaling pathway’s effects on HT-29 cells.

This study aims to explore the anti-proliferation and apoptosis activities of CSO and the mechanism of CSO that leads to HT-29 cell apoptosis by regulating the PI3K/AKT pathway.

## 2. Materials and Methods

### 2.1. Materials and Chemicals

Coix seed was obtained from Liaoyang, Liao Ning Province, China. The CSO was obtained by the ultrasonic-assisted extraction method, using acetone as the extraction solvent, and it was sonicated for 20 min under the conditions of a solid–liquid ratio of 1:20 and an ultrasonic power of 130 W. Standard fatty acid methyl ester (FAME) mixture (#463) was acquired from Nu-Chek Prep Inc. (Elysian, MN, USA). Dulbecco’s modified Eagle’s medium (DMEM) and dimethyl sulfoxide (DMSO) were obtained from Sigma-Aldrich Co. (St Louis, MO, USA). 5-Fluorouracil (5-FU) was obtained from Xudong Haipu Pharmaceutical Co., Ltd. (Shanghai, China). Trypsin-EDTA was purchased from Beyotime Institute of Biotechnology (Shanghai, China). A cell cycle staining kit was bought from MultiSciences Biotech Co., Ltd. (Hangzhou, China). Phosphate-buffered saline (pH: 7.2–7.4) (PBS) and Hoechst 33258 solution were bought from Solarbio Science and Technology Co., Ltd. (Beijing, China). An annexin V-FITC apoptosis detection kit was obtained from the Beyotime Institute of Biotechnology (Shanghai, China). Fetal bovine serum (FBS) was bought from Wisent Inc. (Montreal, QC, Canada). Cell Counting Kit-8 (CCK-8) was obtained from Dojindo Molecular Technologies, Inc. (Kyushu, Japan). Primary antibodies of Bax, Bcl-2, Caspase-3, PI3K, p-AKT^473^, Cyclin-B1, and GAPDH were obtained from Abcam (Abcam, Cambridge, UK).

### 2.2. Characterization of Coix Seed Oil

#### 2.2.1. Fatty Acid Determination

The methyl esterification of CSO followed the method of Hu et al. [29]. Fatty acids were determined by gas chromatography with a flame ionization detector (GC-FID), according to the AOCS protocol [30].

#### 2.2.2. Fourier Transform Infrared (FT-IR) Analysis

ATR-FTIR (Nicolet iS50, Thermo Fisher, Waltham, MA, USA) was used to analyze the functional groups of the CSO. The spectra were generated in absorption mode in mid-IR (ca. 4000–525 cm^−1^) at a resolution of 4 cm^−1^ in 16 scans. The measurement spectrum was analyzed by OMNIC data analysis software, and the transmittance was converted into absorbance.

### 2.3. Cell Culture

HT-29, Caco-2, and HCT-116 cells were obtained from the Shanghai Institute of Cell Resource Center Life Science (Shanghai, China). The three kinds of cells were maintained at a temperature of 37 °C with a 5% CO_2_ atmosphere in DMEM supplied with 10% (volume/volume) FBS and antibiotics (100 U/mL of penicillin and 100 mg/mL of streptomycin) [31]. When the confluence reached 85%, the cells were digested with 0.25% trypsin, collected, and sub-cultured into well plates or Petri dishes according to the needs of the experiment [32].

### 2.4. Cell Viability

The cell inhibition rate was determined according to the instructions of CCK-8 [33]. Briefly, HT-29, Caco-2, and HCT-116 cells (1 × 10^4^ cells/well) were plated into each well of the 96-well plate overnight at 37 °C. Since CSO is insoluble in DMEM, Tween80 was used as an emulsifier. Then, 100 μL/well of different CSO concentrations was added. Tween80 (0.5%) was dissolved in DMEM as a negative control, with 5 μL/mL 5-FU as a positive control. The oil concentration was set to 0.125–10 mg/mL for 24, 48, and 72 h. The blank group was treated with DMEM alone. The cells were washed twice with PBS and cultured with 100 μL DMEM (containing 10 μL of CCK-8 solution) for 60 min. A microplate reader (Bio-Rad Laboratories, Hercules, CA, USA) was used to note the absorbance value at 450 nm. The cell viability of blank cells treated with DMEM only was 100%. The results are presented as a percentage of control. Cell proliferation inhibition of 50% (IC_50_) was calculated using SPSS 22.0 software (IBM Corporation. Armonk, NY, USA).

### 2.5. Cell Morphological Assessment

The cells were seeded in a 6-well plate (2 mL/well) and cultured at a density of 1 × 10^5^/mL overnight. The supernatant was removed, the cells were washed twice with PBS, and then CSO (0.5, 1.0, and 4.0 mg/mL) was added. After the cells were incubated for 24 h, the cell morphology was observed under an inverted optical microscope (CarlZeiss, Jena, Germany) at 40× [34], and 5 μL/mL 5-FU was taken as a positive control.

### 2.6. Hoechst 33258 Staining

Hoechst 33258 staining was used to detect the cell nuclear fragmentation of HT-29 cells. HT-29 cells (2.5 × 10^4^ cells/mL) were cultured in a 24-well cell culture plate. After 24 h of incubation, the cells were treated with 0.5, 1.0, and 4.0 mg/mL CSO and then incubated for another 24 h; 5 μL/mL 5-FU was used as a positive control. After incubation, the cells were fixed with 4% formaldehyde at room temperature and stained with Hoechst 33258 (10 µg/mL) solutions [35]. The stained cells were observed under a fluorescence microscope (Olympus BX 51 TRF, Tokyo, Japan).

### 2.7. Cell Cycle Analysis

HT-29 cells (1 × 10^6^/mL) were cultured in 6-well plates and treated with 0.5, 1.0, and 4.0 mg/mL CSO for 24 h, and 5 μL/mL 5-FU was used as a positive control. Then, the cells were collected by centrifugation and washed with PBS. Then, 1 mL of DNA staining solution and 10 μL of permeation solution were used to fix the cells in a dark room for 30 min [36]. Finally, the cell samples were analyzed by flow cytometry analysis (FACS) (Accuri Cytometers, Inc., Ann Arbor, MI, USA). Cell cycle distribution was analyzed using ModFit software (Verity Software House, Topsham, ME, USA).

### 2.8. Cell Apoptosis Analysis

HT-29 cells (1 × 10^6^/mL) were cultured in 6-well plates and treated with 0.5, 1.0, and 4.0 mg/mL CSO for 24 h, and 5 μL/mL 5-FU was used as a positive control. Then, the cells were stained with annexin V-FITC/PI according to the manufacturer’s instructions [37]. The samples were analyzed by flow cytometry. The data were analyzed using BD FACS Diva software (version. 8.0, Franklin Lake, NJ, USA).

### 2.9. Real-Time PCR (RT-PCR)

HT-29 cells (1 × 10^6^ cells) were treated with CSO (4.0 mg/mL) and incubated for 24 h. After treatment, total RNA was isolated using TRIzol (Invitrogen, Carlsbad, CA, USA) reagent. Then, a prime script first-strand cDNA synthesis kit (TaKaRa-Bio, Dalian, China) was used to convert total RNA into cDNA as a template. Quantitative PCR was performed on the Mx3000P real-time PCR system (Stratagene; Agilent Technologies, Inc., Santa Clara, CA, USA) using SYBR Green PCR master mix.

The primer sequences of PI3K, AKT, Bcl-2, cyclin B1, Bax, caspase 3, and β-actin are shown as follows:

PI3K,

Forward: 5′-CACTGTGGTTGAATTGGGAGA-3′,

Reverse: 5′-CGATTGACAGACAACCATAAGG-3′;

AKT,

Forward: 5′-GAAGACCTTTTGCGGCACAC-3′,

Reverse: 5′-TGTAGAAGGGCAGGCGACC-3′;

Bcl-2,

Forward: 5′-AGGGACGGGGTGAACTGG-3′,

Reverse: 5′-CTACCCAGCCTCCGTTATCC-3′;

Cyclin B1,

Forward: 5′-CCTATTTTGGTTGATACTGCCTCTC-3′,

Reverse: 5′-CTCCATCTTCTGCATCCACATC-3′;

Bax,

Forward: 5′-GGATGCGTCCACCAAGAAG-3′,

Reverse: 5′-TGAAGTTGCCGTCAGAAAACA-3′;

Caspase 3,

Forward: 5′-AGAACTGGACTGTGGCATTGAG-3′,

Reverse: 5′-GCACAAAGCGACTGGATGAAC-3′;

β-actin,

Forward: 5′-TGACGTGGACATCCGCAAAG-3′,

Reverse: 5′-CTGGAAGGTGGACAGCGAGG-3′.

The data were analyzed using the 2^−ΔΔCt^ method. Using β-actin as a reference, the relative mRNA expression level in the sample was calculated, and the results were characterized by the ratio of gene expression to the blank [38].

### 2.10. Western Blot

HT-29 cells were seeded on a 6-well culture plate of 1 × 10^5^ cells·mL^−1^ and allowed to adhere for 24 h. Then, the medium was aspirated, and HT-29 cells were treated with 4.0 mg/mL CSO for 24 h. Refer to the method of Bie et al. to obtain protein bands [39]. Image J software (version. 1.8.0.112, Bethesda, MD, USA) was used to detect the grayscale values of the target bands. The grayscale values of the target protein and the loading control were considered the relative expression of the target protein. GAPDH was used as the internal reference.

### 2.11. Statistical Analysis

All results are expressed as the mean value ± standard deviation (SD) from triplicate tests and were analyzed by IBM SPSS Statistics 22.0 (IBM Corporation. Armonk, NY, USA). All graphics were generated using Origin9.2 program (Origin Lab Corporation, Northampton, MA, USA) software.

From Table 1, it can be seen that 85.10% of coix seed oil was unsaturated fatty acids, which included 51.15% of monounsaturated (MUFA) and 33.95% of polyunsaturated (PUFA) fatty acids. Fatty acids are the main constituents of CSO and play an important role in determining the nutritional value and physicochemical characteristics of the oil. Among these unsaturated fatty acid molecules, oleic acid (C18:1 _cis9_, 50.54 ± 0.63%) was found to be the most predominant. The other types were linolenic acid (C18:2 _cis9,12_), gadoleic acid (C20:1 _cis11_), α-linolenic acid (C18:3 _cis9,12,15_), and palmitoleic acid (C16:1 _cis9_). Among the saturated fatty acids, the most prominent was found to be palmitic acid (C16:0, 11.74 ± 0.50%), followed by stearic acid (C18:0), myristic acid (C14:0), caprylic acid (C8: 0), and arachidic acid (C20:0). Moreover, CSO was found to contain OCFAs, which possess favorable beneficial physiological effects. Therefore, it has numerous potential applications in pharmaceutical and nutraceutical industries.

## 3. Results

### 3.1. Characterization of Coix Seed Oil

Figure 1 shows the peaks at 2980–2800 cm^−1^, which correspond to the C-H stretching of the methyl and methylene backbones of CSO [40]. The peak at 1740 cm^−1^ is attributable to the C=O carbonyl stretching of lipid and fatty acid ester groups, which correspond to the total lipids in the oil. C-H scissoring was found at 1460 cm^−1^, and the C-O stretching vibration of ester groups was observed at 1150 cm^−1^ [41]. Additionally, the spectra exhibited a peak associated with the cis C=C out-of-plane bending at 720 cm^−1^ [42]. The FT-IR spectrum of CSO matched well with the vegetable oil band stretching given by Rohman and CheMan [43].

### 3.2. CSO Inhibited Cell Growth in HT-29, Caco-2, and HCT-116 Human Colon Cancer Cells

As shown in Figure 2, CSO has a dose-dependent and time-dependent inhibitory effect on HT-29, Caco-2, and HCT-116 cells. With the extension of time, the IC_50_ of colon cells gradually decreased. The lowest IC_50_ values for CSO against Caco-2, HCT-116, and HT-29 cells were 2.84 ± 0.21, 3.00 ± 0.25, and 1.78 ± 0.11 mg/mL for 72 h, respectively (Table 2). The results showed that the inhibitory effect was the most obvious after 72 h. Comparing the IC_50_ values of different cells at the same time, the IC_50_ values of HT-29 cells were the lowest. The IC_50_ values for CSO against Caco-2, HCT-116, and HT-29 cells were 7.00 ± 0.34, 8.74 ± 0.17, and 5.30 ± 0.21 mg/mL, respectively, for 24 h. It can be seen that CSO could exert the same inhibitory effect at a lower concentration for HT-29 cells. The results demonstrated that CSO presented restriction activity on HT-29 cells in a time- and dose-dependent manner. Similarly, in the previous study, CSO was also able to de-adhere to HCC cells and restrain their anti-proliferation activity [35]. Therefore, the HT-29 cells were selected as subjects in order to investigate the tumor-suppressive effect of CSO.

### 3.3. Morphology Observations

The cell morphology of HT-29 cells treated with CSO is shown in Figure 3a. The untreated cells showed a normal, healthy shape, uniform size, and clear structure. The cells treated with 5-FU showed obvious shrinkage and changes in cell morphology. CSO-treated cells showed abnormal morphology and decreased adhesion, and some cells became rounded. The cell line presented a dose-dependent inhibitory effect. The higher the concentration, the more obvious the changes in cell morphology. These results demonstrate that CSO may motivate HT-29 cell apoptosis. A similar influence against HT-29 cell morphologic changes was found when HT-29 cells were treated with a cordycepin fraction extracted from *C. militaris* [44].

### 3.4. Apoptotic Cell Observations by Hoechst 33258 Staining

In HT-29 cells treated with CSO (0.5, 1.0, and 4.0 mg/mL), a dose-dependent induction of apoptosis could be observed by staining with Hoechst 33258 dye. As presented in Figure 3b, normal and apoptotic cells displayed a weak blue and strong blue fluorescence, respectively. The results demonstrated that the apoptosis rate of the CSO group gradually increased with increases in CSO concentration. After 24 h of incubation with the CSO treatment, apoptotic cells were identified as those with condensed nuclei, indicated by a strong blue color. In the blank group, there were almost no strong blue colors, indicating that some, but relatively few, of the cells were apoptotic. After treatment with different concentrations of CSO, as the concentration increased, the strong blue color gradually increased, showing the dose-dependent inhibitory effect of CSO, which led to cell apoptosis.

### 3.5. The Cell Cycle Distribution of HT-29 Cells

The results of cell cycle arrest are shown in Figure 4. The results showed that the increase in G_2_ cells in HT-29 cells treated with CSO proceeded in a concentration-dependent manner after 24 h. HT-29 cells exposed to CSO at 0.5, 1.0, and 4.0 mg/mL clearly exhibited cell death, showing 8.84 ± 0.04%, 10.33 ± 0.58%, and 17.33 ± 1.15% increase in the G_2_ phase, while showing 6.19 ± 0.20% in the blank. After exposure to 5-FU at 5 µg/mL, a significant increase of 24.60 ± 0.78% in the S cell population in HT-29 cells was recorded, as compared to the blank (16.56 ± 0.46%) (*p* < 0.05). Overall, these data suggest that CSO induces HT-29 cell cycle arrest in the G_2_ phase. This cell cycle arrest was also observed in HSC-3 cells in the presence of cinnamon essential oil [45].

### 3.6. CSO Induced Apoptosis in HT-29 Cells

The results of apoptosis by flow cytometry are shown in Figure 5. Flow cytometry data precisely showed that CSO initiated cell death by activating early and late apoptotic processes. The staining results showed that more than 96% of the cells were alive in the blank; the rates for the early and late apoptotic stages were 1.57% and 1.50%, respectively. HT-29 cells exposed to 0.5, 1.0, and 4.0 mg/mL of CSO induced early and late apoptosis, with increases of 6.67%, 5.03%, and 10.73% recorded in the lower right quadrant and 11.33%, 20.67%, and 23.70% recorded in the upper right quadrant, respectively. This quantitative measurement suggests that the CSO induced most of the cells into late apoptosis. This is in concordance with an earlier report that found that CSO induces apoptosis [35].

### 3.7. Gene Expression Levels

As shown in Figure 6, CSO induced cells with two pro-apoptotic genes (caspase 3, Bax) and one anti-apoptotic gene (Bcl-2), showing different expression patterns. HT-29 cells treated with CSO showed a significant increase in caspase-3 and Bax mRNA expression. In addition, CSO also significantly down-regulated the expression of PI3K, Akt, and Bcl-2 mRNA (*p* < 0.05). Up-regulation of caspase-3 is essential for inducing cell apoptosis. CSO exposure led to a 1.45-fold increase in caspase-3 mRNA expression compared to the blank group. Caspase-3 is frequently stimulated to catalyze the death of apoptotic cells. Therefore, it is a precise indicator of apoptosis. Assessing Bax mRNA expression in CSO incubation indicated a fold-change value of 1.33. This corresponded to the character of Bax and Bcl-2 genes in apoptosis, which can shift the balance to cell proliferation or death [46]. Additionally, the Bax protein conceals the effects of Bcl-2, leading to increased apoptosis [46]. CSO showed that the expression of Bcl-2 mRNA, which matched the blank group, decreased by 0.76 times. According to earlier reports, the PI3K/Akt signal transduction pathway allows cells to control transmission, which is a meaningful element in suppressing cell apoptosis and pushing cell growth and reproduction [47]. CSO revealed a 0.81- and 0.73-fold decrease in PI3K and Akt mRNA expression, respectively, compared with the blank group. Cyclin B_1_ plays an important role in regulating the cell cycle [48]. Assessment of Cyclin B1 mRNA expression in CSO incubations revealed a fold-change value of 1.17. This was coordinated with the ability of CSO to monitor the Bax and Bcl-2 proteins and mRNA expression, which is conducive to HCC apoptosis [35].

### 3.8. Western Blot Analysis

As depicted in Figure 7a,b, the HT-29 cells treated with 4.0 mg/mL of CSO for 24 h showed a significant increase in the apoptotic protein expression of Bax and caspase-3 (*p* < 0.05) and a significant decrease in the anti-apoptotic protein expression of Bcl-2 compared to the blank group (*p* < 0.05). The above results indicated that CSO induced apoptosis of HT-29 colon cancer cells related to Bax and Bcl-2. After CSO treatment, the expression of CyclinB_1_ in HT-29 cells was up-regulated. Cell cycle and apoptosis genes regulate the behavior of the corresponding proteins in cells, which, in turn, leads to cell cycle arrest and even apoptosis. Furthermore, the PI3K/Akt signal transduction pathway’s protein expression was down-regulated in HT-29 cells. PI3K and p-AKT^473^ were significantly decreased (*p* < 0.05). These data together imply that CSO might inhibit AKT with PI3K inhibition.

## 4. Discussion

CRC is the most common malignant tumor of the gastrointestinal tract, and its conventional treatments have certain side effects [49]. Therefore, natural plant ingredients have been studied for their ability to reduce the side effects of conventional treatments and to improve the survival rate of cancer patients [50]. Coix seeds have long been used in traditional Chinese medicine (TCM), and many researchers have suggested that CSO exhibits tumor-suppressive activity [14].

We characterized the fatty acid content and functional groups of coix seed oil, and the results showed that coix seed oil mainly contains oleic acid (50.54%), linoleic acid (33.76%), palmitic acid (11.74%), and stearic acid (2.45%), and the fatty acid functional groups matched well with the vegetable oil band. Coix seed oil exhibited anti-cancer activity in T24 cancer cell lines; this was attributed to palmitic acid and linoleic acid in an appropriate ratio to oleic acid [21]. In the study, the ratio was 0.90, which was close to 1. The results indicated that CSO can inhibit the growth of cancer cells.

In this study, we demonstrated that coix seed oil (CSO) could significantly induce the apoptosis of HT-29 cells. Apoptosis is a cell process that is essential for organ development, tissue remodeling, and immune regulation, and it plays an important role in regulating cell growth and tissue homeostasis [51]. The results of a CCK-8 assay demonstrated that CSO could inhibit the proliferation of HT-29 cells in a dose- and time-dependent manner; an IC_50_ value (50% inhibition) of 5.30 mg/mL was recorded at 24 h. In addition, Figure 3 shows the changes in the morphology and nucleus of HT-29 cells after CSO treatment. As shown in Figure 4 and Figure 5, the results revealed that CSO-treated HT-29 cells provided higher cell numbers in the G_2_ phase, which demonstrated CSO-induced cell cycle arrest. This might have been caused by the feasibility of DNA damage and the collapse of repair mechanisms in the cells.

The PI3K/Akt pathway is an important pathway that regulates protein synthesis, cell cycle progression, proliferation, apoptosis, and cytokine stimulation [52,53]. Akt activated by PI3K is the main mediator of signal transduction in this pathway, and the amount of phosphorylated AKT influences the proliferation and survival of cells [54,55]. In order to study the ability of coix seed oil to regulate the PI3K/Akt signaling pathway in HT-29 colon cancer cells, the mRNA and protein levels of PI3K and Akt were detected by RT-PCR and Western blot analysis. The results showed that the expression levels of PI3K and p-Akt^473^ protein after CSO treatment were significantly lower than those of the blank group.

Bcl-2 is the downstream molecule of Akt in the PI3K/Akt pathway. It mainly plays a role in anti-apoptosis, which is involved in mediating cell survival, proliferation, and translation regulation [56]. Bax can neutralize the effect of Bcl-2 and cause the activation of downstream apoptosis signaling pathways [57]. Caspase-3 has attracted much attention for its role as the executor of cell apoptosis. The results of this study showed that the expression levels of Bax and caspase-3 increased while the expression level of Bcl-2 decreased [38].

Taken together, the regulation effect of CSO on the apoptosis pathway of HT-29 cells is displayed in Figure 8. Hence, suppression of the PI3K/Akt pathway may represent a potential anti-cancer therapeutic direction for CSO. The study confirms that CSO is an effective, novel, and low-toxicity natural agent for the treatment of colon cancer. These results expand current knowledge of how CSO acts as a potent adjuvant with tumor-suppressive activity.

## 5. Conclusions

This work indicates that CSO has an anti-proliferative effect on HT-29 in vitro and that it can induce apoptosis and cell cycle arrest. Coix seed oil mainly contains oleic acid, linoleic acid, palmitic acid, and stearic acid and has the characteristic functional groups of vegetable oils. On the one hand, this suppression effect of CSO can be attributed to cell cycle arrest at the G_2_ phase, which increases the expression of Cyclin B_1_. On the other hand, the decrease in PI3K/AKT activation is related to CSO administration. Further, CSO can lead to cancer cell apoptosis through activating caspase-3, up-regulating Bax, and downregulating Bcl-2. This study provides new insights into the possible molecular mechanisms of CSO-induced apoptosis and should lead to the increased recognition of CSO in terms of its importance in healthy food and agricultural applications.

## Figures and Tables

**Figure 1 foods-10-02833-f001:**
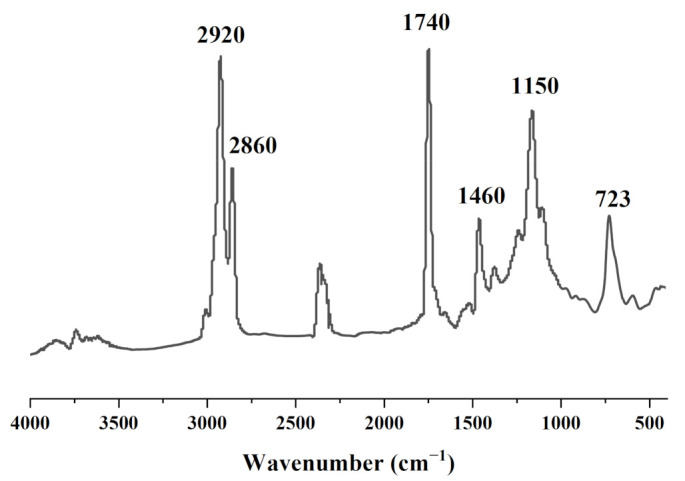
FTIR spectra of coix seed oil.

**Figure 2 foods-10-02833-f002:**
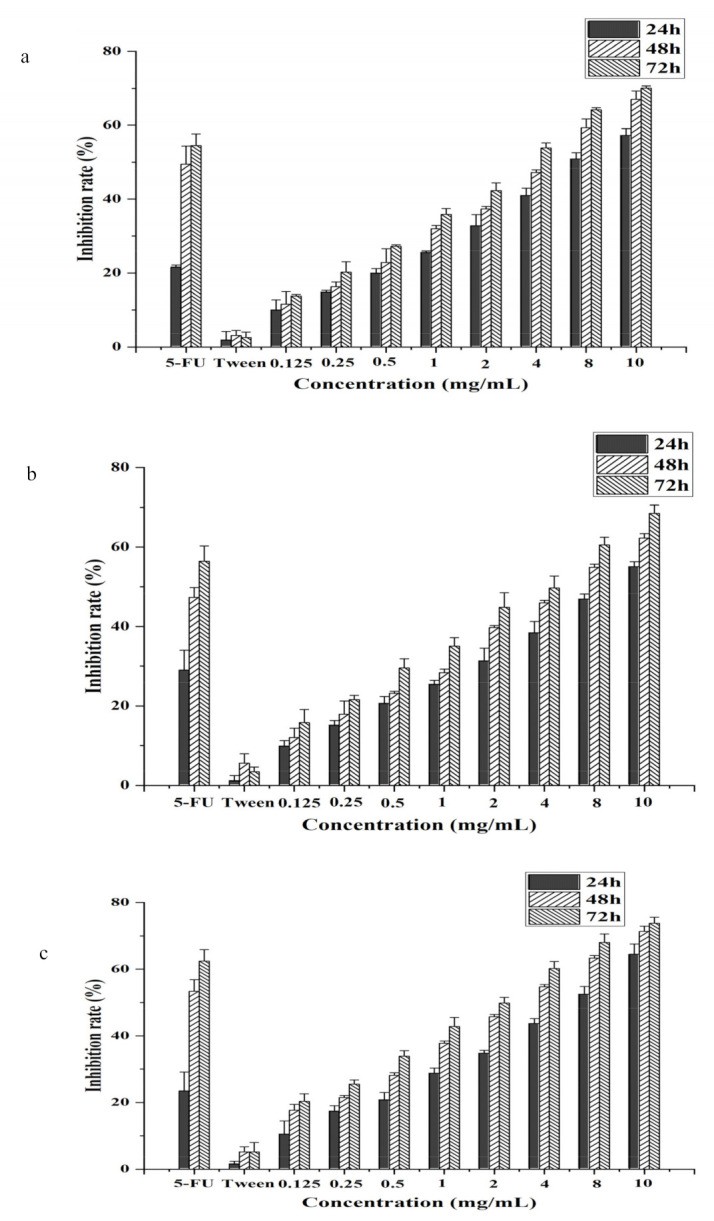
(**a**) Effect of CSO on Caco-2 cells. (**b**) Effect of CSO on HCT-116 cells. (**c**) Effect of CSO on HT-29 cells.

**Figure 3 foods-10-02833-f003:**
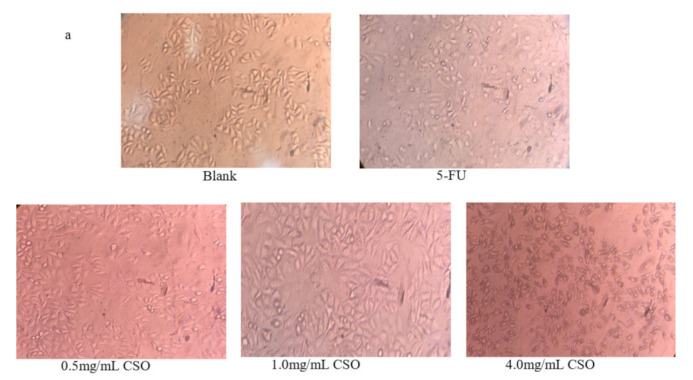
(**a**) Cell morphology was observed by inverted light microscopy (⊆ 40) after HT-29 cells were treated with CSO (0.5, 2.0, and 4 mg/mL) for 24 h. (**b**) CSO-induced apoptosis in HT-29 cells using Hoechst 33258 staining by fluorescent microscope (⊆ 200) after cells were treated with CSO (0.5, 2.0, and 4 mg/mL) for 24 h. Abbreviations: 5-FU, 5-fluorouracil; CSO, coix seed oil; apoptotic bodies are indicated by red arrows.

**Figure 4 foods-10-02833-f004:**
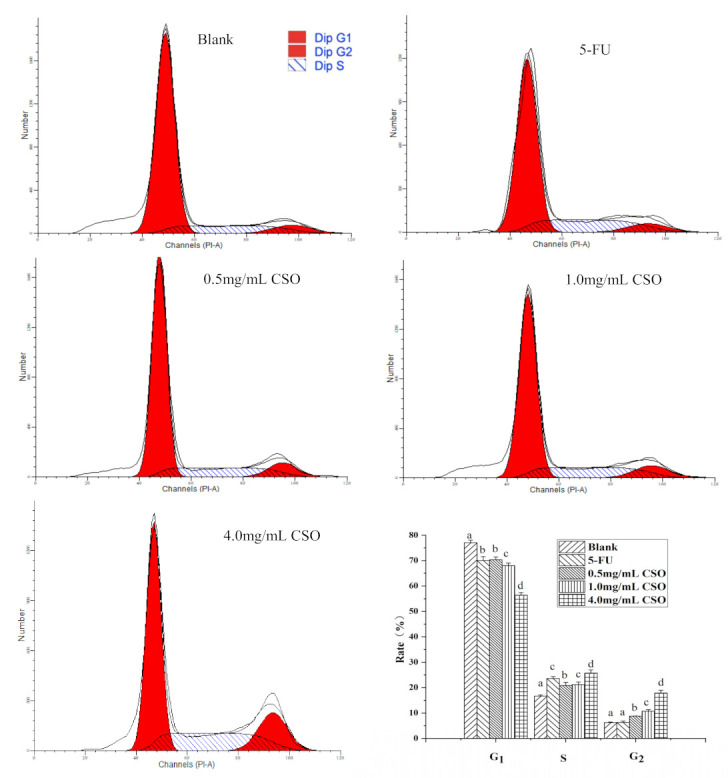
Cell cycle distributions of HT-29 cells in the absence and presence of CSO. Different letters indicate significant differences (*p* < 0.05) according to Duncan’s test. Abbreviations: 5-FU, 5-fluorouracil; CSO, coix seed oil.

**Figure 5 foods-10-02833-f005:**
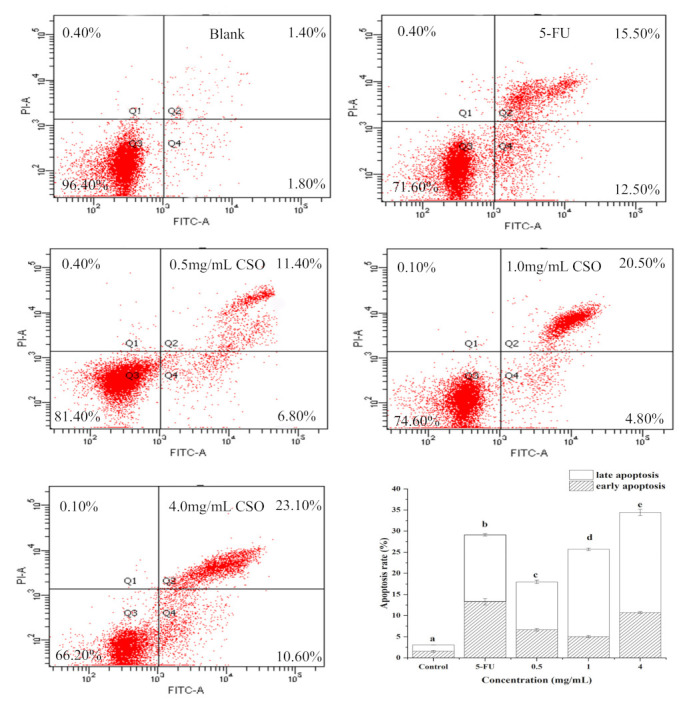
Cell apoptosis of HT-29 cells in the absence and presence of CSO. Different letters indicate significant differences (*p* < 0.05) according to Duncan’s test. Abbreviations: 5-FU, 5-fluorouracil; CSO, coix seed oil.

**Figure 6 foods-10-02833-f006:**
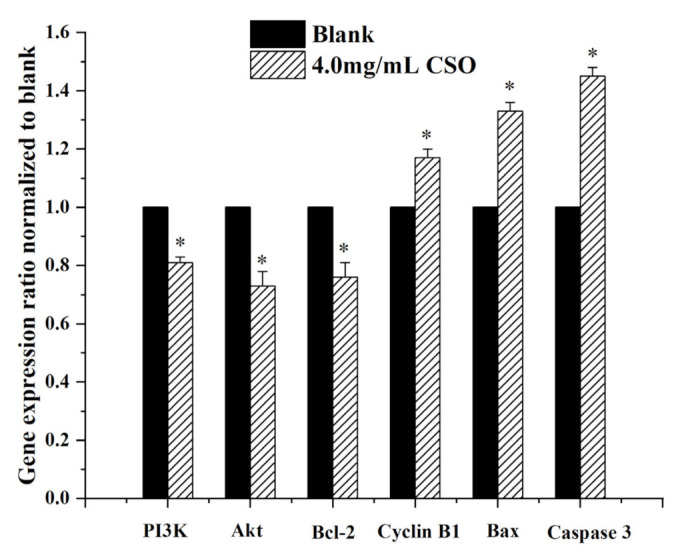
Impact of CSO on PI3K, AKT, Bcl-2, cyclin B1, Bax, and caspase-3 in HT-29 cells; * *p* < 0.05, vs. the blank.

**Figure 7 foods-10-02833-f007:**
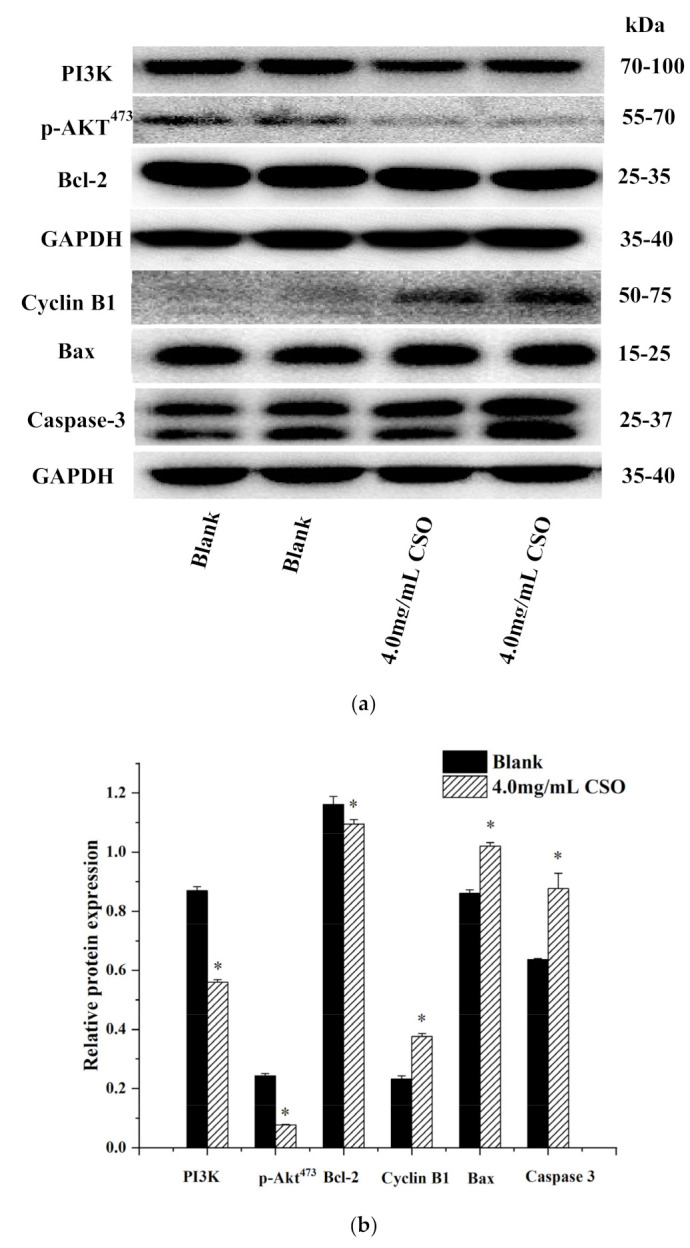
(**a**) Protein bands of PI3K, p-AKT^473^, Bcl-2, cyclin B1, Bax, and caspase-3, determined by Western blot; (**b**) protein expression of PI3K, AKT, Bcl-2, cyclin B1, Bax, and caspase 3 determined by Western blot; * *p* < 0.05, vs. the blank.

**Figure 8 foods-10-02833-f008:**
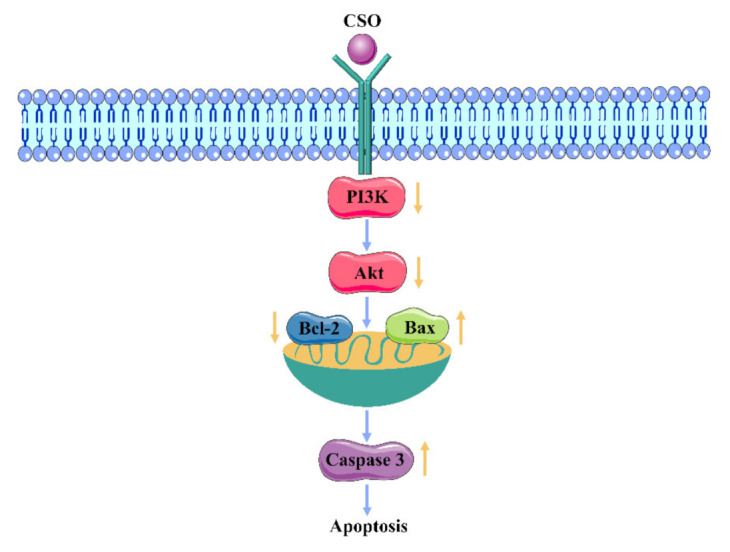
Model of the mechanism of CSO action on HT-29 cells by regulating the PI3K/AKT pathway.

**Table 1 foods-10-02833-t001:** Fatty acid composition (g/100 g) of CSO.

Fatty Acid Composition	CSO
Saturated fatty acid	14.46 ± 0.48
Caprylic acid (C_8:0_)	0.07 ± 0.01
Myristic acid (C_14:0_)	0.17 ± 0.01
Palmitic acid (C_16:0_)	11.74 ± 0.5
Stearic acid (C_18:0_)	2.45 ± 0.06
Arachidic acid (C_20:0_)	0.03 ± 0.01
Monounsaturated fatty acid	51.15 ± 0.73
Palmitoleic acid (C_16:1 cis9_)	0.16 ± 0.04
Oleic acid (C_18:1 cis9_)	50.54 ± 0.63
Arachidonic acid (C_20:1 cis11_)	0.45 ± 0.06
Polyunsaturated fatty acid	33.95 ± 0.07
Linoleic acid (C_18:2 cis9,12_) (ω-6)	33.76 ± 0.21
α-linolenic acid(C_18:3 cis9,12,15_)(ω-3)	0.19 ± 0.01
Odd carbon fatty acid	0.44 ± 0.11
Undecanoic acid (C_11:0_)	0.02 ± 0.00
10-Heptadecenoicacid (C_17:1 cis10_)	0.04 ± 0.01
Heptadecanoic acid (C_17:0_)	0.09 ± 0.03
Tricosanoicacid(C_23:0_)	0.29 ± 0.05
(C_16:0_ + C_18:2_)/C_18:1_	0.90
Unsaturated fatty acid/Saturated fatty acid	5.88

**Table 2 foods-10-02833-t002:** IC_50_ values of CSO for 24, 48, and 72 h on Caco-2, HCT-116, and HT-29 cells.

	IC_50_(mg/mL)		
	24 h	48 h	72 h
Caco-2	7.00 ± 0.34 ^a^	4.00 ± 0.21 ^a^	2.84 ± 0.21 ^a^
HCT-116	8.74 ± 0.17 ^b^	4.80 ± 0.22 ^b^	3.00 ± 025 ^a^
HT-29	5.30 ± 0.21 ^c^	2.56 ± 0.30 ^c^	1.78 ± 0.11 ^b^

Different letters on the same column indicate significant differences (*p* < 0.05) according to Duncan’s test.

## Data Availability

Data supporting reported results are available upon request.

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
