# Peer review of "Anti-Cancer Properties of Coix Seed Oil against HT-29 Colon Cells through Regulation of the PI3K/AKT Signaling Pathway"

_foods, 2021, doi:10.3390/foods10112833_

Round 1
Reviewer 1 Report
Review for Foods
Anti-cancer Properties of Coix Seed Oil Against HT-29 Colon Cells Through Regulation of PI3K/AKT Signaling Pathway
By Chunlei Ni et al.
General considerations
The present work studies the anticancer capacity of the oil obtained from the seeds of a species of grass native to East Asia and Malaysia, known as Moses' tear and whose food and medicinal use is widespread throughout East Asia.
The authors use, in their study, the product marketed as Coix Seed Oil (CSO) investigating its anticancer properties in three colon cancer lines, HT-29, Caco-2 and HCT-116. All three with different genetic-molecular characteristics. The first, dependent on p53, the second, deficient in p53, and the third, mutated in the KRAS proto-oncogene.
When we analyze the methodological development used by the authors, we have to say that it is appropriate for these studies, since cytotoxicity (CI) is studied at 24, 48 and 72 hours; changes in cell morphology by inverted light microscopy and Hoechst analysis by fluorescence microscopy; changes in the phases of the cell cycle; changes in apoptosis by flow cytometry; as well as changes in the expression of mRNA levels and marker proteins of this anticancer activity. All this, within the classic approach of this type of research work.
However, and although the authors finally demonstrate what they propose in their working hypothesis, there are many aspects, of greater consideration, that must be corrected, answered and carried out new experiments.
Major objections
1.- Title: The title does not fully show the content of the work presented, since the authors only mention the anticancer capacity in HT-29 cells when other cell lines are also studied in the manuscript (Caco-2 and HCT -116). The authors should focus their work on two lines HT-29 and Caco-2 as they are antagonistic in p53, and, therefore, carry out the experiments corresponding to both lines. All of this would lead to an appropriate modification of the title.
2.- Introduction: The introduction is relatively short, especially regarding the PI3K/AKT signaling route since the authors base their work on it. Therefore, the authors should add new comments related to this intracellular signaling pathway, both in its meaning and in its use in other works, especially those related to the dual behavior of this pathway and later use these new citations in the discussion comparing the results.
3.- Results:
- a) It is absolutely essential that the authors include, either in this section or in the material and methods section, the qualitative and quantitative composition of the phytochemicals present in the CSO samples used in the study. In some way, the anticancer activity should be identified with the major phytochemical components of the oil.
- b) The biggest problem that I observe in the present work is the IC-50 values found by the authors in the different cancer lines. The values found, for example, at 48 hours for HT-29 and Caco-2 range between 2.56 and 4.00 mg · mL-1, and these are values of an order 50 to 100 times higher than those that would correspond to those of compounds with standard anticancer activity. In addition, the authors use a dose, practically twice the value of IC-50 for the rest of the experiments (4.0 compared to the 2.56 mg · mL-1, typical of HT-29). The authors should explain why they do that.
- c) Studies of gene expression by PCR (RT-PCR): The values presented by the authors in Figure 5 represent the relative amounts of mRNA expression of different proteins used in this study and using 4 mg · mL-1 of CSO. The values represent the times the values change in relation to the absence of CSO. This figure should be completed and include the real values that have allowed the results to be relativized.
- d) Western blot analysis. The results in Figure 6 represent the expression changes of different marker proteins. In it, only a control band is observed for all of them. Have not the authors made a control for each protein? If not, why? On the other hand, the authors use a control protein, GADPH, which does not exist in cells. On the contrary, there is GAPDH (Gliceradehyde-3-phosphate dehydrogenase). Clearly the authors should correct this error and, furthermore, specify the use of this control protein in the appropriate section of Material and methods.
- e) Let's talk about the PI3K/AKT route. PI3K is a kinase that phosphorylates many proteins, including AKT. The phosphorylation of AKT makes it active and as a result increases the proliferation and survival of pathological cells. When PI3K is inhibited by CSO, the amounts of phosphorylated AKT decrease and therefore proliferation and survival are significantly reduced. On the other hand, the amounts of non-phosphorylated AKT decrease, increasing the values of p53 versus phosphorylated p-53, which allows the cellular apoptosis values to increase.
The authors show, indeed, that CSO reduces the expression of PI3K, which coincides with the role that an anticancer compound should have, but they also observe that the expression of AKT decreases, without specifying that it is phosphorylated AKT (active as kinase) or unphosphorylated AKT (not active as kinase), and that must be properly explained as the consequences are completely opposite.
Minor objections
1.- Authors should correct different syntactic errors throughout the manuscript.
2.- Authors should separate the results and discussion sections and that there are enough results to do so.
3.- In accordance with what is stated in the previous sections, the authors must modify the figures once they carry out the new experiments if they consider it appropriate.
Reviewer 2 Report
The manuscript needs significant revision in some experiments and the main text. The following main points should be clarified and explained in the revised version:
- The authors indicated the presence of some majors’ constituents. Could they report the individual activity for each of them on the selected cell lines? And discuss the contribution to the extract activity?
- In figure 2. B: a. The bleu staining also indicated the staining of the nucleus. Can the authors provide a detailed explanation and clear high-magnitude image about the apoptotic bodies in this study? b. Please provide evidence of the presence or absence of nuclear fragmentation. If the authors can reconsider this section of the experiment and provide the appropriate data and discuss them to support the author's finding? c. Please improve the image presentation in Figures 2A and 2B and also include a precise scale bar.
-
Why did the authors not perform a double staining experiment Hoechst 33342/PI?
- In figure 4, If the authors can include the percentage data in each quadrant. Could the authors provide a clear explanation of the gating in the flow cytometry experiment?
- Could the authors provide the complete GC-MS profiling data and consider all the observed constituents in the discussion.
Reviewer 3 Report
In the manuscript entilted „Anti-cancer properties of coix seed oil against HT-29 colon cells through regulation of PI3K/AKT Signaling pathway” Chunlei Ni, et al. described the studies of coix seed oil as an anticancer agent. The authors also made an attempt to study its influence on the cytotoxic activity towards colon cells. There are several issue which I would like the authors to tackle:
- The introduction – it is too long, especially the paragraph on cancer does not bring anything new. Please make it more condensed.
- Correct the paper in terms of language mistakes (eg. line 47, 85 etc.).
- CCK-8 assay measures rather cell growth inhibition, not proliferation.
- What is the effect of CSO on primary cells?
- Unfortunately, there is no discussion.
- The oil should be characterized phytochemically. What are the main identified ingredients and active compounds?
- I am afraid of the fact that the dose ok 4mg/mL is far too much for the studies, and it will hard to obtain such concentration in vivo.
Round 2
Reviewer 3 Report
I have no more comments.
Author Response
Thank you, your previous comments have already enriched my article. Your comments were highly insightful and enabled us to greatly improve the quality of our manuscript.